# Polymeric Nanoparticles Properties and Brain Delivery

**DOI:** 10.3390/pharmaceutics13122045

**Published:** 2021-11-30

**Authors:** Laís Ribovski, Naomi M. Hamelmann, Jos M. J. Paulusse

**Affiliations:** Department of Molecules and Materials, MESA+ Institute for Nanotechnology and TechMed Institute for Health and Biomedical Technologies, Faculty of Science and Technology, University of Twente, P.O. Box 217, 7500 AE Enschede, The Netherlands; l.ribovski@utwente.nl (L.R.); n.m.hamelmann@utwente.nl (N.M.H.)

**Keywords:** nanoparticles, polymers, brain delivery, blood–brain barrier, controlled drug delivery, therapeutics, nanomedicine

## Abstract

Safe and reliable entry to the brain is essential for successful diagnosis and treatment of diseases, but it still poses major challenges. As a result, many therapeutic approaches to treating disorders associated with the central nervous system (CNS) still only show limited success. Nano-sized systems are being explored as drug carriers and show great improvements in the delivery of many therapeutics. The systemic delivery of nanoparticles (NPs) or nanocarriers (NCs) to the brain involves reaching the neurovascular unit (NVU), being transported across the blood–brain barrier, (BBB) and accumulating in the brain. Each of these steps can benefit from specifically controlled properties of NPs. Here, we discuss how brain delivery by NPs can benefit from careful design of the NP properties. Properties such as size, charge, shape, and ligand functionalization are commonly addressed in the literature; however, properties such as ligand density, linker length, avidity, protein corona, and stiffness are insufficiently discussed. This is unfortunate since they present great value against multiple barriers encountered by the NPs before reaching the brain, particularly the BBB. We further highlight important examples utilizing targeting ligands and how functionalization parameters, e.g., ligand density and ligand properties, can affect the success of the nano-based delivery system.

## 1. Introduction

The advent of nanomedicine has brought nanoparticles that provide unique ways to control interactions with cells and tissues. However, when facing the complexity of in vivo systems, nanoparticles are not simply required to interact with one singular cell type, but rather with several cellular environments with distinctive characteristics before reaching the intended target site. For disorders in the NVU, reaching the diseased site is even more complex, as the CNS is well-guarded by its own immune system and a specialized endothelial barrier. The treatment and diagnosis of diseases such as Parkinson’s, Alzheimer’s, and brain cancers becomes challenging, as most molecules cannot reach the brain at therapeutically relevant doses. Nanocarriers pose as an auspicious concept for improving the delivery of therapeutics to the CNS. However, systemically administrated nanocarriers that target the brain must also overcome the mononuclear phagocytic system (MPS) and be transported across the blood–brain barrier (BBB), a highly selective structure [1,2]. Crossing the BBB generally implies that the nanoparticles (NPs) should be internalized by endothelial cells and subsequently exocytosed into the brain (Figure 1). Furthermore, after reaching the brain, the nanocarriers need to get to the different brain areas for effective therapeutic delivery. To obtain systems with the most favorable results, understanding and tuning of nanocarriers properties is essential.

The highly versatile nature of polymers offers exceptional opportunities to carefully modulate NP properties. The variety in composition of polymeric nanoparticles is extensive, e.g., single polymers, copolymers, protein-based, lipid–polymer hybrids, metal–polymer hybrids, etc. [3,4,5,6,7,8,9]. Each of those systems can be adjusted to obtain particles of different sizes, with different release and degradation profiles, possibilities for further functionalization, and many other properties. In drug delivery, the design of particle properties enables better control over particle interactions within the biological environments affecting biodistribution, clearance, transport across barriers, uptake, and ultimately, therapeutic effect. Particle characteristics such as size, surface charge, and targeting ligand coupling are well-known to have an impact on particle interaction with cells and accumulation in tissues, including the brain [10,11,12,13,14,15]. However, one property alone does not define the effectiveness of the delivery, but rather an ensemble of properties will help to better tune the interactions between particles and cells or barriers, highlighting the importance of proper NP design.

In this review we outline the main NP physicochemical properties evaluated for brain delivery and how they benefit particle transport and accumulation, in addition to the effect of targeting ligands on cerebrovascular targeting and tissue targeting (Figure 2). Potential systems for the treatment of brain cancer and neurodegenerative diseases are also discussed.

## 2. Nanoparticle Size

Various properties of NPs influence the in vivo targeting abilities. A key factor is NP size, as the biodistribution and cellular uptake of NP is highly dependent on it [16]. NPs that are intravenously injected, and that therefore evade the gastrointestinal barriers, have been shown to accumulate in different organs based on their size [17,18]. Particles > 200 nm are effectively cleared by the spleen, while smaller particles < 5 nm are rapidly cleared from the body by the kidneys. Therefore, nanoparticles in the range of 10 to 200 nm are most commonly evaluated in nanomedicine. For the application of NPs to cross the BBB, size has been shown to dramatically influence the transport efficiency [19]. In a biodistribution study by De Jong and coworkers, (gold) NPs of 10, 50, 100, and 250 nm in diameter were injected into rats’ tail veins, and the majority of NPs were shown to accumulate in the liver and spleen [18]. However, only the 10 nm size NPs were detected in the brain, indicating that smaller particle sizes are advantageous in crossing the BBB. For polymer NPs, a similar trend has been observed. Methotrexate-loaded polybutylcyanoacrylate (PBCA) NPs with a polysorbate-80 coating of 70, 170, 220, 345 nm were injected into rats; the highest drug delivery was observed for the 70 nm sized NPs [20]. This size-dependent transport was also observed in an animal model, by intravenous injection of NPs into the tail vein of rats. Furthermore, the coated NPs achieved higher drug accumulation in the brain compared with non-coated NPs. Cruz and coworkers used poly(lactic-co-glycolic) (PLGA) NPs functionalized with a polyethylene glycol (PEG) coating and targeting moieties of 800 CW, a NIR dye, to study particle diffusion in mice with traumatic brain injuries [21]. The evaluated NPs had a size range of 100 to 800 nm. Deepest penetration into the brain was observed for the 100 nm NPs with the 800 CW coating. This study shows that NP size also influences transport in models with a compromised BBB. PLGA NPs have been also used in targeting studies using trehalose, where brain distribution in rats was evaluated for particle sizes of 71 and 147 nm [22]. As earlier seen by Cruz and co-workers, the 71 nm NPs presented significantly higher accumulation in the brain. Polystyrene (PS) NPs have been frequently used to investigate size-dependent transport, as a wide size range of PS NPs is commercially available [23,24]. However, the vast variety in evaluated surface functionalizations impede the comparison of particle transport. PS particles with a TPGS (d-α-tocopheryl polyethylene glycol 1000 succinate) coating and uncoated PS NPs of 25, 50, 100, 200, and 500 nm were analyzed in a cellular uptake and biodistribution study [23]. The highest cellular internalization in Caco-2, human colorectal adenocarcinoma cells, and Madin-Darby Canine Kidney (MDCK) cells was observed for 100 nm NPs; however, this was only significant for TPGS-coated NPs in MDCK cells. Even though Caco-2 and MDCK cells are not brain endothelial cells, they are epithelial cells that also form cell barriers, and certain parallels can be drawn for the particles’ behavior in polarized brain endothelial cell barriers. Subsequently, in the biodistribution study, NP accumulation in the brain was correlated with their size, showing decreased accumulation with increasing particle size.

However, size-dependent transport of NPs has also been contradicted by some studies. To improve the prediction of particle transport, Zhang and co-workers worked on a mathematical model that includes particle size and surface charges [24]. In the range of 20 to 200 nm, they found ~150 nm to be the optimum size for neutral and positive charged NPs, but they also state that the optimum size varies depending on the types of cells and conditions. In an in vitro study with PS NPs comparing particles sizes of 100, 200, and 500 nm, the highest permeability was observed for the 200 nm NPs [25]. Voigt et al. studied PBCA NPs with various surfactants and demonstrated that the surfactant had a greater influence on particle transport than the size of the NPs [26]. Overall, the trend of size-dependent transport across the BBB has been confirmed for a wide variety of polymeric NPs, though other particle parameters are also key in the process.

## 3. Nanoparticle Shape

A wide variety of strategies has been developed for the preparation of polymeric nanoparticles, but typically spherically shaped NPs are obtained. Methods to obtain anisotropic polymeric particles, e.g., rods, discs, and worms (Figure 3), are lithography, controlled crosslinking or polymerization, film-stretching, and microfluidics approaches; however, those are not yet straightforward and still limit particle application [27,28,29,30,31,32,33,34,35,36,37]. Studies with polystyrene nanorods decorated with targeting moieties indicate that rod-shaped particles display stronger adhesion and increased internalization by brain endothelial cells under both static and flow conditions, in addition to higher brain accumulation in vivo [25,31,38,39]. Nanorods were functionalized with anti-intercellular adhesion molecule 1 (ICAM-mAb), anti-ovalbumin (OVA-mAb), and immunoglobulin G (IgG) antibodies to evaluate the interaction with brain endothelium and for biodistribution studies with anti-transferrin receptor (TfR-mAb), anti-ICAM-mAb, and IgG [31,39]. For non-specific interactions (observed for IgG-particles), the spherically shaped particles displayed stronger adhesion under flow conditions, suggesting a relation between specific interactions and shape. Another study employing vascular adhesion molecule-1 (VCAM-1) and IgG gave similar results, highlighting that under static conditions, uptake between IgG- and VCAM-1-coated particles were distinguishable, with a greater increase for VCAM-1-functionalized rod-shaped particles of up to 2.5-fold, though under flow conditions, only the elongated particles presented distinct uptake levels—a 1.5-fold increase [15]. Mendes and collaborators evaluated biodistribution, clearance, and circulation of radiolabeled PEG-b-polystyrene micelles by single-photon emission computed tomography (SPECT), showing no significant differences in brain accumulation for spherical and elongated particles [40]. It supports the role of specific interactions in shape-dependent behavior; however, the authors also point to different stiffness observed in the center of the elongated particles, which may also affect their behavior.

A shape-dependent interaction in an Alzheimer’s disease (AD)-related environment was studied using high-density lipoprotein (HDL) particles [41]. HDL particles are composed of lipids and associated proteins, including apolipoprotein A-I (Apo I), and can be shaped as spherical and discoidal particles. In vitro and mathematical models suggest that Apo I-associated discoidal HDL particles are transported more efficiently across the BBB, diminish β-sheet concentration on β-amyloid fibrils, and induce enhanced structural destabilization of β-amyloid plaques as compared with the spherical particles.

As pointed out in many in vitro and in vivo studies, anisotropic particles are internalized by macrophages to a lesser extent than spherical particles and, consequently, have a longer circulation time, which may favor accumulation in the brain [30,42].

NP anisotropy influences particle adhesion and avidity. Studies indicate that ligand-functionalized NPs that induce specific interactions may profit from anisotropy and that enhanced targeting may ensue [15,31]. Still, the impact of different shapes of polymeric NPs in brain delivery requires more investigation, which is currently limited by the lack of well-established and broadly applicable methodologies for preparing such NPs.

## 4. Nanoparticle Stiffness

Nanoparticle stiffness was largely overlooked for many years as a property to regulate delivery. However, its relevance has been emerging over the past 5 years and offers new delivery concepts, including in brain delivery. Particle stiffness may be altered by crosslinking density, composition, and structure. Crosslinking density is controlled through the addition of different amounts of crosslinking agent during nanoparticle preparation [43,44]. Structural variation of stiffness is often approached by the preparation of core-shell structures in which core stiffness is changed, ranging from hollow shells to solid cores [45,46]. Composition can affect stiffness in different ways. Thermoresponsive nanogels, for example, shift from a swollen state to a condensed state upon increasing temperature, which can occur above (or below if condensed to swollen) the volume phase transition temperature (VPTT), while VPTT depends on the monomer. For example, poly(N-isopropylacrylamide (p(NIPAM)) nano/microgels, a widely studied type of nanogel, have a lower critical solution temperature (LCST) around 32 °C, while poly(N-isopropylmethacrylamide) (p(NIPMAM)) nano/microgels have an LCST around 44 °C. Under physiological conditions, 37 °C, such particles will present different behavior, with p(NIPAM) being a collapsed particle and p(NIPMAM) a swollen particle [43,47].

Although many aspects remain to be understood, current reports on the effects of stiffness on brain delivery already provide interesting insights. Anselmo et al. evaluated the biodistribution behavior of PEG diacrylate (PEGDA)-based hydrogel nanoparticles, with stiffnesses ranging from 10 to 3000 kPa (diameter around 200 nm), in mice via intravenous injection [48]. Brain accumulation was more pronounced for soft particles at 30 min; however, at the 12 h time point, these differences were no longer significant. Other organs also showed higher retention, e.g., heart and lungs, for soft particles. The authors attribute the differences in accumulation to the particles’ circulation time. Soft particles had prolonged circulation as compared with hard particles, with elimination half-life in 6-fold higher for the soft particles, which manifested in higher accumulation in organs with elevated blood flow. In another report, PEGDA nanogels were compared with carboxylated polystyrene NPs in vitro under brain flow conditions [25,49]. A higher association with the endothelial cells was observed for the polystyrene particles (stiff), as well as increased transport rates across the endothelial monolayer. PEG-block-poly(pentafluorophenyl methacrylate) polymers were crosslinked with different diamine cross-linkers, giving polymersomes with tunable membrane elasticity (from 3.7 to 7.3 MPa) that were evaluated against an orthotopic glioblastoma (GBM) tumor model [50]. Lower elasticity led to increased transcytosis and brain accumulation, despite shorter blood circulation. It should be highlighted that the surface of the particles might play an important role in the outcome since PEG is well known for its ability to reduce protein adsorption and avoid internalization. Nonetheless, several studies indicate that stiffer polymeric nanoparticles are indeed internalized to a greater extent by brain endothelial cells, yet it does not guarantee higher transcytosis. Employing a filter-free BBB model and p(NIPMAM) nanogels with different crosslinking densities, Zuhorn and coworkers showed that lower stiffness nanogels (<200 kPa) promote transcytosis even though endocytosis is not promoted [44]. Brown et al. also attempted to investigate the effect of stiffness on the BBB with protein, liposomes, and polystyrene nanoparticles; however, analyses could not be dissociated from other particle properties, such as size, composition, and shape [10]. The influence of stiffness on transcytosis and brain accumulation adds a parameter to the design of NPs that may be evaluated for polymeric NPs. Comparison should always observe the stiffness range, often described by the particles’ Young’s moduli and not simply as softer or stiffer.

Simulation of the interaction between a lipid membrane and particles of varying stiffness suggests that to initiate and complete the wrapping process of a soft particle more energy is required than in the case of stiffer particles. In particular, membrane bending deformation is more pronounced for stiff particles, meaning the softer particles will require an additional energy barrier to induce deformation in the membrane to be fully wrapped [51,52,53,54].

## 5. Nanoparticle Surface Characteristics

### 5.1. Surface Charge

Surface chemistry dictates the fate of NPs in vivo, and a straightforward but important characteristic is surface charge. To induce surface charges on NPs, functional groups such as carboxylic acid or amines are frequently utilized [55,56,57]. Neutral particles have been shown to circulate longer in the bloodstream and avoid rapid clearance, as compared with charged particles [58]. Positively charged NPs typically display higher accumulation in the spleen and liver, as compared with negatively charged NPs. NP surface charges likewise also impact transport across the BBB. The endothelial cells of the BBB have a higher density of negative charges due to the presence of proteoglycans, as compared with blood components and human umbilical vein endothelial cells (HUVEC) [59]. This leads to the ability of positively charged NPs to cross the BBB via adsorptive-mediated transcytosis [60]. Moscariello et al. observed BBB crossing for NPs with cationic surface coatings [55,56]. A 5 nm dendrimer protein bioconjugate was formed with biotin-functionalized cationic poly(amido)amine (PAMAM) dendrons of second and third generation, demonstrating transport across the BBB without impairing the BBB integrity and entering NVU cells [56]. However, positive surface charges on NPs can also induce toxicity by increasing the production of reactive oxygen species, which leads to mitochondrial damage. Furthermore, positive surface charges, as well as strong anionic charges, have even been shown to decrease BBB integrity [61]. This may result in higher particle transport, but at the price of BBB disruption potentially inciting future health effects in patients, which can lead to higher transport due to the disruption of the BBB. In a study on polystyrene NPs, uptake was compared with the transport across a Caco-2 cell barrier, presenting higher uptake for NPs with positive surface charges, as well as higher transport for negative NPs [57]. Therefore, careful control over the surface charge is key for inducing NP transport across the BBB and limiting NP accumulation in endothelial cells, while also controlling the cytotoxicity of NPs.

### 5.2. Ligand Density and Linker Length

Ligands are utilized in active cellular targeting, where highly specific interactions with cell receptors are exploited. Based on the targeting of viruses, multivalent targeting has been studied and shown to increase the binding affinities of viruses to the cell surfaces, increasing internalization [62]. However, higher ligand density on NP surfaces may also cause side effects, such as size increases, steric hindrance, and decreased stealth behavior in vivo [62]. Therefore, controlling cellular uptake of NPs in relation to ligand density has sparked the interest of the nanomedicine field. Research focusing on linker length, specifically studying PEG linkers and shorter peptide linkers, revealed higher cellular internalization with shorter linker length (Figure 4) [63,64]. Ligand density was optimal at 100% surface functionalization, as studied by Abstiens and coworkers for RGD ligands in combination with PEG linkers (M_w_ = 2 kDa) in glioblastoma cells [63]. The effective surface coverage of the ligands is likely considerably lower than 100% due to the length of the PEG linker. The interaction of NPs and cells was analyzed via a mathematical modeling approach, focusing on ligand density in relation to receptor density [65,66]. The optimal ligand density for highest cellular internalization was found to be dependent on receptor density. However, as ligand density increases, the rise in binding affinity may, in effect, decrease the exocytosis efficiency [65]. An optimum balance is key for effective transport across the BBB. Anraku and coworkers studied the effect of glucose density on the surface of micelles that interact with glucose transporters type 1 (GLUT-1) on brain endothelial cells [12]. They showed that the polymeric micelles successfully crossed the BBB and that micelles with 25% surface functionalization showed the highest accumulation in the brain. Depending on ligand density, the particles were either found in and around the endothelial cells when high ligand densities were employed, or they penetrated deeper into the brain in neurons and microglia at lower ligand densities, confirming the importance of carefully balancing the binding affinity of NPs. Ligand density has a substantial impact on the avidity of the NPs, which ultimately affects their uptake, intracellular trafficking, and transcytosis, which is discussed to a greater extent in Section 5.4, Avidity.

### 5.3. Targeting Ligands

Ligand-decorated nanoparticles are widely explored for brain delivery to promote transcytosis and targetability. To boost nanoparticle transport across the BBB, ligands that bind cell receptors are a common design feature of nanoparticles [67,68]. Often targeting overexpressed receptors or transporters at the apical surface of polarized brain endothelial cells, the decorated NPs typically display increased uptake by the cells, which results in higher transcytosis levels. Once the particles cross the BBB, it is still critical that they also accumulate at the appropriate location in the brain. Targeting can be improved by additional binding moieties or by controlling ligand density. Receptor recruitment during endocytosis can favor but also limit internalization and transcytosis of NPs. Ligand-associated properties, e.g., ligand affinity and ligand density, may saturate the receptors and reduce NP uptake or hinder NP release from the basolateral membrane [12,65,69,70]. Dual targeting can benefit the transport of NPs to the brain as well as accumulation in the desired site. Research in dual targeting of endothelial cells has shown increased NP crossing using a combination of ligands, avoiding saturation of receptor recruitment [71,72]. Moreover, dual or multifunctional ligands can tackle barriers and achieve site-specific targeting in one system. Examples of dual targeting strategies are discussed in this section.

Targeting strategies are, however, susceptible to a number of parameters. Proper targeting design goes beyond the choice of effective ligands and requires control over conjugation parameters, e.g., ligand density and linker nature [12,70,73], in addition to understanding particle surface interactions with complex biological media [74,75]. Common conjugation strategies are based on covalent binding to surface groups or polymer chains, e.g., amine-carboxyl or thiol-maleimide conjugations. Moreover, covalent conjugation, electrostatic adsorption, and non-covalent receptor–ligand-mediated conjugation are also frequently employed [76,77].

#### 5.3.1. Glucose and Glucose Derivatives—Glucose Transporters

Many neurological disorders are associated with variations in the expression and activity of glucose transporters at the blood–brain barrier or in diseased cells, e.g., Alzheimer’s disease and cancer [78,79,80]. Glucose transporters type 1 (GLUT1) are generally identified as being responsible for glucose uptake by brain endothelial cells, even though other glucose transporters are also present on the cell surface, e.g., GLUT3 and GLUT4 [81,82]. In most cancers, increased expression of glucose transporters is observed. For example, in glioblastoma, the necrotic areas exhibit upregulation of GLUT1, as compared with invasive borders [83]. Therefore, glucose transporters have been explored to enhance BBB transport, tumor penetration, brain accumulation, and treatment efficacy [12,84,85]. For example, poly(ethylene glycol)-co-poly(trimethylene carbonate) NPs functionalized with 2-deoxy-d-glucose were developed as a system for exploring the expression profile of glucose transporters in the BBB and glioma cells [84]. This targeting approach increased the treatment efficacy of paclitaxel-loaded particles. As previously discussed, ligand density may have a significant role in the interaction between NPs and cells. Anraku et al. demonstrated that polymeric micelles decorated with varying densities of glucose directed the accumulation of the micelles to neurons, in addition to modulating the transport across the BBB [12]. Immunohistochemical analysis revealed that a fraction of 25% of glucose conjugated through the C6 position increases brain accumulation and does not remain significantly associated with the brain capillary endothelial cells, in contrast with micelles with 50% of functionalized block copolymers (Figure 5), which can be explained by the affinity of the micelles to the cellular surface. The fractions of 25% and 50% glucose were primarily located in the neurons and microglia. Surprisingly, astrocytes did not present detectable levels of the micelles [12]. Kataoka and coworkers described above addresses in a comprehensive manner how relevant control over ligand density is for NP-mediated brain delivery, distinguishing association levels of micelles to different components of the NVU. Antisense oligonucleotide (ASO) therapies have also been demonstrated to benefit from a controlled density of glucose moieties on the nanocarriers [86]. Polyion micelles of the copolymer PEG-b-poly(l-lysine) modified with 3-mercaptopropyl amine and 2-thiolaneimine (PEG-PLL(MPA/IM)) at various glucose amounts (ratio glucose/methoxy terminal groups: 0, 24, 52, 76, 103) displayed enhanced brain accumulation for micelles, with a ratio of glucose ligands of 52 and a knockdown efficiency that correlates with brain accumulation.

The addition of a second type of targeting ligand can be advantageous. This was demonstrated for example by Wu et al. who combined maltobionic acid, a glucose derivative, with 4-carboxyphenylboronic acid, a sialic acid binding moiety, as ligand on cholic acid-functional dendrimers in a sequential manner, resulting in enhanced targeting [85]. A reduction in tumor growth, as well as an increase in survival rate in mice with orthotopic patient-derived xenografts (PDX) of diffuse intrinsic pontine glioma (DIPG), were observed upon administration of vincristine-loaded nanocarriers. The results were correlated with a synergistic contribution of increased particle transport across the BBB, owing to the interaction with glucose transporters, and higher uptake and deeper penetration in tumor spheroids. MRI and near infra-red fluorescence (NIRF) imaging in vivo and ex vivo support brain tumor accumulation of the nanocarriers.

#### 5.3.2. Transferrin, Anti-Transferrin Antibodies, and Transferrin Receptor Targeting Peptides—Transferrin Receptor

Transferrin receptor (TfR) is a well-known target for diagnosis and treatment of cancers. The transferrin–transferrin receptor complex (Tf-TfR) is involved in iron metabolism, i.e., from iron capture and transport by Tf, to internalization in cells by complexation with TfR, followed by degradation or exocytosis. The BBB is rich in TfRs, which have been extensively explored for improving transcytosis, and consequently, for improving brain delivery. Transferrin-decorated NPs (Tf-NPs) are commonly tested for their ability to overcome the BBB and accumulate in the brain [87,88,89]. Even though Tf-NPs often show amelioration of transcytosis levels, it should be taken into account that in vivo, the presence of endogenous Tf will compete for TfR sites and will ultimately limit the availability of receptors for Tf-NPs. Bearing this in mind, employing an antibody as ligand that does not compete for the same site as the endogenous Tf may be beneficial. Antibody-conjugated PEGylated liposomes show a significant improvement over Tf-conjugated counterparts, though the binding mode of the antibodies may affect the final outcome [90,91]. For example, particle avidity towards receptors can also negatively affect transcytosis levels; this is discussed in another section. A range of antibody clones has been reported in the literature for TfR, e.g., OX26 [92,93], Ri7, RI7217 [94], 8D3 [95].

Using two-photon excitation, Kucharz and coworkers traced the delivery of anti-TfR-functionalized PEGylated liposomes to the brain in anesthetized and awake mice [14]. Post-capillary venules are responsible for nearly all delivery, even though high association was observed in the capillaries. Evidence of transcytosis of nanoliposomes and their presence in the brain is presented, greatly contributing to the discussion of nanoparticles’ capability to cross the BBB.

TfR can also be further targeted using peptides. T7-peptide, for example, combined with Tet1 peptide, was conjugated to dendrigraft poly-l-lysine NPs, where Tf was conjugated via acid-cleavable long polyethylene glycol to decouple from the NPs upon entry in the endo/lysosomes, augmenting transcytosis levels [96]. The Tet1 targeting peptide also further allowed neuron targeting.

#### 5.3.3. Peptides

Phage display technology has been successfully used to identify new peptides to improve brain delivery [97]. Through screening of a library of peptides by affinity, the consensus motifs are identified through the clones, and the most promising peptides can be tested. The use of peptides avoids competition with endogenous ligands, allows design versatility, and reduces costs. In brain delivery, BBB-shuttle peptides are often associated with receptors or transporters, as previously described for TfR, but adsorptive transcytosis and passive diffusion have also been documented [98].

##### G23 Peptide

Also known as Tet1 peptide, the G23 peptide consists of 12 amino acids (HLNILSTLWKYR or HLNILSTLWKYRC with C-terminal cysteine for coupling) and has been shown to enhance the transcytotic capacity of NPs across the BBB [13,99]. Although derived from tetanus toxin, a powerful neurotoxin, the peptide lies within a heavy chain, which on itself is non-toxic and displays neuron-targeting capabilities with binding properties to gangliosides [100,101]. Polymersomes prepared from PEG-block-poly(caprolactone-gradient-trimethylene carbonate) were decorated with G23 peptide and shown to increase transcytosis by at least 4-fold as compared with non-targeted polymersomes in in vitro models [99,101,102]. Significant in vivo brain accumulation was also reported [102].

Moreover, the peptide has been demonstrated to be an effective ligand across a range of brain-related diseases. NPs made of zein, an alcohol-soluble protein from maize, carrying curcumin were functionalized with G23 peptide and revealed not only higher transcytosis levels but also antitumor activity and deeper penetration in tumor spheroids of glioblastoma [13]. In Alzheimer’s disease models, it offers advantages in BBB transport and targeting of neurons, with promising results of decorated nanocarriers decreasing amyloid plaque [96,103,104]. Aiming to reduce the expression of β-site amyloid precursor protein cleaving enzyme 1 (BACE1) through small interfering RNA (siRNA) technology, siRNA was loaded into PEGylated poly(2-(N,N-dimethylaminoethyl)methacrylate (PEG-PDMAEMA) nanocarriers functionalized with a CGN peptide (d-CGNHPHLAKYNGT), which also promotes transcytosis, together with Tet1 peptide [104]. This nanotherapeutic was able to decrease BACE1 mRNA levels and promote neuroprotective effects, which was reflected in improvements in cognitive performance in amyloid precursor protein/presenilin 1 (APP/PS1) double transgenic mice.

##### Angiopep-2, Apolipoprotein E, and Other Low-Density Lipoprotein Receptor-Related Protein 1(LRP1) Ligands—LRP1 Receptor

Low-density lipoprotein receptor-related protein 1 (LRP1) is a cell surface receptor expressed in the BBB, brain cancers, and neurons [105,106]. Multiple ligands can bind to LRP1, including apolipoprotein E (Apo E) and lactoferrin [107,108]. Apo E is implicated in the maintenance of the integrity of the BBB and fibrillogenesis in Alzheimer’s disease [109]. Adsorption or coupling of Apo E ligands on the surface of polymeric NPs shows improvement in transport across the BBB, accumulation in the brain and tumors, and inhibition of amyloid plaque accumulation [110]. Apo E is also associated with the protein corona and with the effectiveness of brain delivery nanocarriers that favor Apo E occurrence in the corona composition [111,112].

A well-established transport peptide that binds to LRP1 is angiopep-2 (TFFYGGSRGKRNNFKTEEY), which has been extensively explored in brain delivery [113,114,115,116]. Hyaluronic acid NPs conjugated to angiopep-2 were developed as a theranostic platform for glioblastoma [117]. Uptake of NPs equipped with this ligand by glioblastoma cells was superior to non-targeted NPs, while the encapsulated gadolinium-diethylenetriamine imaging agent endowed the NPs with diagnostic properties. Nanoconjugates of β-poly(L-malic acid) were functionalized with angiopep-2 and tri-leucine, and their delivery to the brain was compared with nanoconjugates where angiopep-2 was substituted with other ligands, e.g., transferrin receptor ligands [118]. The reported results suggest that tri-leucine increases brain uptake in areas with high cerebral microvasculature density, e.g., midbrain colliculi. Angiopep-2 was also shown to significantly contribute to the transport across the BBB, where other ligands showed reduced transcytosis.

##### RGD Peptide

Arg-Gly-Asp or RGD is a highly conserved motif associated with cell adhesion to the extracellular matrix [119,120]. As an integrin-binding motif, RGD is involved in the regulation of many processes, e.g., angiogenesis, cell migration, metastasis, and apoptosis, which makes the RGD-integrin pair of interest as a target. The targeting capabilities and therapeutic benefits against brain cancer of cyclic(RGD) peptides were demonstrated for PEG-b-poly-(L-glutamic acid) micelles incorporating (1,2-diaminocyclohexane) platinum(II) [121]. Coadministration of micelles functionalized with cyclic(RGD) or cyclic(RAD), cyclic-Arg-Ala-Asp, revealed the progression of transport through the BBB and tumor accumulation of the cyclic(RGD)-micelles observed by intravital confocal laser scanning microscopy. The RGD peptide is an effective ligand for siRNA therapy. Lou et al. prepared polyplexes via self-assembly employing N-(2-hydroxypropyl)-methacrylamide (HPMA) and N-acryloxysuccinimide (NAS) random copolymers synthesized by RAFT polymerization with siRNA stabilized by cholesterol [122]. The polyplexes were PEGylated, and cyclic(RGD) peptides were coupled onto the PEG chains. RGD’s binding ability increased the internalization of the polyplexes in human glioblastoma cells and successfully silenced gene expression. In a dual targeting approach aimed at superior targeting ability and delivery to tumor neovasculature and tumor cells, PEG-PCL nanoparticles were functionalized with RGD peptide and interleukin-13 peptide [123]. Ex vivo imaging of treated mice indicated great accumulation of the dual-functionalized NPs in the tumor site in the brain; also, deeper penetration in tumor spheroids was observed for the dual-functionalized particles.

##### Glutathione

Glutathione (GSH), γ-l-glutamyl-l-cysteinyl-glycine, is an important antioxidant produced in cells. PEGylated liposomes display improved brain delivery when they are coated with GSH [124,125,126]. By encapsulating antibody fragments into GSH-PEGylated-liposomes, Rotman et al. argue that such liposomes are promising tools for the targeted delivery of drugs, as they can cross the BBB and enhance brain accumulation [126]. As a gene delivery-mediating targeting ligand, GSH is also a suitable moiety. GSH moieties were attached to a poly(ethylene imine) derivative post-polymerization; moreover, the ratio of primary and secondary amine groups was varied [127]. BBB transport studies in hCMEC/D3 cells revealed that primary amines display stronger association with and increased uptake in the endothelial monolayer; however, they are not prone to transport through the BBB, likely due to intracellular pathways.

##### RVG Peptide

Derived from the rabies virus, which can reach the brain through peripheral nerves and enter the central nervous systems via retrograde axoplasmic transport, RVG peptides have been successfully employed as targeting ligands in delivery to the brain. While the rabies virus does not reach the brain through transport across the BBB, the RVG peptides do show potential as BBB-penetrating ligands in several drug delivery systems [128,129,130,131]. RVG peptides were conjugated to PEG-b-PLGA nanoparticles loaded with 4,4′-dimethoxychalcone (DMC), a molecule that promotes autophagy [132]. The therapeutic effect was evaluated for Parkinson’s disease using a 1-methyl-4-phenyl-1,2,3,6-tetrahydropyridine (MPTP)-induced Parkinson’s disease mouse model [133]. The results suggest that enhanced transport across the BBB takes place, along with a reduction in oxidative stress and inhibition of tyrosine hydroxylase ubiquitination, as well as improvement in motor deficits [132]. Cook et al. studied the distribution of RVG29-modified PLGA nanoparticles in the brain [134]. Major particle accumulation occurred in the cortex and cerebellum, likely due to cerebral blood volume, for both targeted and non-targeted NPs. Accumulation of targeted NPs was increased compared with non-targeted NPs; however, the effect was not sustained over 6 h after injection. Loss of targeting ability was attributed to instability of the RVG29 peptide in aqueous solutions. RVG peptides have also been reported as potential carriers for gene therapies to the brain [135,136].

##### TGN Peptide

The TGN peptide (TGNYKALHPHNG) was identified by phage display screening, with the consensus of clones of 60% [137]. It has been reported to boost the brain transport efficiency of PEG-PLGA, PEG-PDMAEMA, and PEG-poly(ε-caprolactone) (PCL) [104,137,138,139,140]. By encapsulating a neuroprotective peptide in PEG-poly(lactic acid) (PEG-PLA) NPs and decorating these particles with a fusion peptide of TGN and Tet1 bound through a tetraglycine linker, therapeutic efficacy was improved as compared with non-decorated and single-peptide-decorated NPs. NPs functionalized with the fusion peptide exhibited superior targeting to BBB, as evidenced by the higher association with brain capillary endothelial cells (bEnd.3 cells) and increased brain accumulation determined by ex vivo fluorescence imaging [139].

#### 5.3.4. Aptamers

Aptamers consist of single-stranded DNA or RNA and spontaneously fold in 3D structures that bind to a specific molecule. Aptamer-based systems for brain delivery mainly focus on ways to target the disease site [141,142,143,144,145]. A technique often used to identify aptamers is SELEX, which stands for systematic evolution of ligands by exponential enrichment [146,147]. Monaco and coworkers showed that PLGA-b-PEG NPs conjugated to aptamer Gint4.T, which interacts with platelet-derived growth factor receptor β in glioblastoma, show significantly higher tumor accumulation than NPs conjugated with a scrambled version of the aptamer [148]. Transport across the BBB into damaged and non-damaged areas was also higher for the Gint4.T-modified NPs. Aptamers targeting CD133 have also been reported to improve brain delivery in a multi-drug polymer–micellar system composed of poly(styrene-b-ethylene oxide) (PS-b-PEO) and PLGA [149].

#### 5.3.5. Cell Membrane Coating

The selection of binding ligands is an intricate task, especially considering the complexity and variability of cells, barriers, and microenvironments. Each cell membrane has a specific composition that comprises several proteins, carbohydrates, and lipids. As an approach to interacting more precisely with specific cells, biomimetic nanoparticles have been developed by coating NPs with various types of cell membrane [150,151,152,153,154]. Cancer cell membranes are particularly interesting due to the homotypic adhesion between cells within the tumor microenvironment and their ability to camouflage themselves from the MPS, owing to the presence of anti-phagocytic signs, such as CD47 [155,156]. Employing metastatic brain tumor cells, poly(ε-caprolactone) (PCL) NPs loaded with indocyanine green were coated with these cell membranes and evaluated in brain tumors [157]. The authors not only reported increased penetration across an intact as well as a disrupted BBB and brain accumulation but also reported superior treatment efficacy. Cancer cell membranes were also employed in the treatment of ischemic stroke by coating PEG-block-poly(2-diisopropyl)methacrylate (PEG–PDPA) NPs loaded with succinobucol with 4T1 cell membrane, a metastatic breast cancer cell line. Targeting of ischemic inflammation lesions was achieved, and greater BBB penetration was observed [158].

### 5.4. Avidity

Another property that is of great relevance to targeting is avidity, the combination of abilities to bind receptors. It is particularly relevant for transport across the BBB, where the nanocarriers should complete the apical-to-basolateral transport to reach the brain. Related to the antibodies–antigen complex stability, avidity regulation may avoid that the antibodies and antibody-decorated NPs remain in the microvasculature, but rather allow them to be delivered to the brain parenchyma. A series of anti-TfR antibodies with different affinities was prepared by introducing alanine mutations in order to determine whether their lower or higher affinities could impact transcytosis [69]. The antibodies with higher affinities showed greater uptake by the endothelial cells; however, the antibodies of lower affinity displayed increased brain accumulation. The authors discuss the idea that high affinity promotes uptake, as more receptors are likely to be bound to cell surface TfR, though low affinity antibodies are more prone to subsequently dissociate from the TfR. Similar behavior was observed for glucose transporter targeting and low-density lipoprotein receptor-related protein 1 (LPR1) targeting [12,73]. By functionalizing poly[oligo(ethylene glycol) methacrylate]–poly [2(diisopropylamino)ethyl methacrylate] with angiopep-2, which binds LPR1, it has been shown that syndapin-2 proteins are involved in transcytosis and that this process depends on the avidity of the carrier [159]. Carriers with high affinity promote faster degradation, while mid-affinity carriers promote fast shuttling, with formation of tubular structures associated with syndapin-2. The formation of sorting tubules for degradation and exocytosis has also been described, exploring TfR comparing mono- and bivalent anti-TfR monoclonal antibodies. The effect of particle avidity on associating with the sorting tubules during intracellular trafficking sheds light on a remarkable sorting mechanism and factors that may control it. It not only reinforces the importance of NP design in improving brain delivery but also points out that controlled ligand functionalization is beneficial for targeted drug delivery.

Simulations of clathrin-mediated endocytosis and exocytosis, in which avidity of rigid nanoparticles is included, indicate that a critical affinity needs to be reached to allow complete internalization and that too high affinity impedes particle release from the basal membrane due to insufficiently large fusion pore formation [65]. Additionally, too high ligand density will refrain particle detachment from the membrane, even though pore size is enough for particle release (Figure 6). The simulation aims to predict the behavior of NPs with different avidities through receptor-mediated transcytosis, an essential process for the transport of NPs across the BBB to reach the brain.

### 5.5. Protein Corona

Protein corona formation is an intrinsic property of nanoparticles that are exposed to biological environments and is often modulated by a nanoparticle’s physicochemical properties [160,161,162,163]. Corona composition is dynamic, and protein mobility plays an important role in the desorption/adsorption process. The Vroman effect addresses protein corona formation in materials exposed to a physiological environment and postulates that protein exchange will change corona composition, while the overall quantity of proteins remains roughly the same [163,164]. As reported for many cellular types, the uptake levels and mechanisms of internalization of nanoparticles are affected by the composition of this protein corona [165,166,167,168]. Particle biodistribution and clearance are also significantly influenced by the protein corona. Proteins such as serum albumin, apolipoproteins, fibrinogen, clusterin, and Ig gamma and light chains are commonly found in the protein corona of nanoparticles [163]. Many of these proteins are relatively featureless in terms of targeting; however, some are well-described ligands that cross the BBB via receptor-mediated transcytosis, e.g., Apo E. There have been promising attempts of controlling the corona formation to improve brain delivery of NPs. For example, Zhang et al. reported the functionalization of PEG-liposomes with a short peptide that binds to the lipid-binding domain of apolipoproteins, leaving the receptor-binding domain available to interact with lipoprotein receptor-related proteins and scavenger receptors class B, which affected intracranial glioma accumulation and notable increases in brain distribution [169]. Even though the protein corona is of interest in brain delivery, it also poses challenges in the investigation of particle properties [170,171,172].

Coating nanoparticles with molecules that avoid protein adsorption is a broadly implemented strategy in nanotherapeutic systems, and PEGylation is the most common approach. Increased circulation times and brain accumulation are reported for PEG-coated NPs prepared from polycyanoacrylate [173,174,175], poly (lactic-co-glycolic acid) [176,177], polystyrene [176], and other polymers [178], which is associated with the decreased formation of protein corona and evasion of the MPS. PEG density regulates corona formation and, consequently, particle clearance, of which the threshold for rapid clearance is indicated as 20 PEG chains per 100 nm^2^ for PLGA-PG nanoparticles, largely independent of particle size (diameters of 55, 90, and 140 nm) [177]. No distinguishable difference regarding the composition of the protein corona was observed, although Apo E amounts decrease with increasing PEG density. Moreover, 100 nm diameter polystyrene NPs coated with nine PEG chains per 100 nm^2^ showed good diffusion in rat brain tissue ex vivo, as determined by nanoparticle tracking analysis, as compared with particles with lower densities. Even below the PEG density threshold reported by Bertrand et al., brain diffusion appears to benefit from increased PEG density, which may be attributed to the higher quantity of Apo E in the protein corona and reduced adhesiveness [176,177]. Other surface modification methods to control protein corona formation are surface charge and hydrophobicity modulation [179,180,181].

Particles’ mechanical properties also result in different protein corona profiles, which may affect uptake, toxicity, transport across the BBB, and brain accumulation [44,48,182,183]. Protein adsorption is lower for softer particles, which ultimately leads to a reduced impact of corona-related particle–cell interactions [183,184].

Targeting capabilities of ligand-decorated NPs are likewise also affected by the presence of a protein corona [75]. Corona formation and the effects on targeting have been evaluated for PEGylated polystyrene NPs functionalized with Tf [74]. Particles were submitted to in vitro and in vivo protein corona formation conditions, respectively, by incubating particles in human plasma and by injecting the particles in the tail vein of male Kunming mice. Particles were purified prior to targeting experiments. Both types of coronae diminished transport across the BBB for the Tf-NPs to a level comparable with non-targeted NPs. Tf-NPs with a protein corona formed in vivo did, however, still present some targeting ability, as represented by higher internalization levels. Tumor cell targeting, however, remained for both corona types, although it was higher for NPs of which the corona was formed in vitro.

We should highlight that corona formation is not restricted to the presence of proteins and that other biomolecules, e.g., sugars, are also present and may also affect particle behavior [168].

In Table 1, we summarize the effects of NP properties in brain delivery and the main advantages and limitations of those properties. We should also highlight that loading of molecules to the NPs affects the properties, e.g., size and stiffness, that loaded NPs should be considered as a distinctive NP, and that characterization should be performed accordingly.

## 6. Concluding Remarks

Systemic administration of nanoparticles provides a promising and effective approach towards therapeutics delivery to the brain. The many different synthetic pathways towards polymeric nanoparticles enable careful design of new nanocarriers. The modular fashion in which desired features can be incorporated allows exquisite control over physicochemical properties, which can effectively contribute to overcoming barriers and can help to direct particle interactions with cellular/biological environments. The blood–brain barrier and the mononuclear phagocytic system are critical barriers to deliberate in order to accomplish successful delivery and tissue accumulation. The influences of some nanoparticle properties have been documented extensively—in particular, nanoparticle size, where smaller nanoparticles favor transport across the BBB and brain accumulation, provided that they are not too small (<5 nm) and can avoid clearance by the kidneys. Other nanoparticle properties that affect cell–nanoparticle interactions are still underexplored. Ligand density and stiffness are excellent examples. Their influence on biodistribution, transcytosis, and brain accumulation have been addressed and demonstrated to be relevant parameters, though most literature reports on brain delivery do not characterize or optimize nanocarriers regarding these parameters. The examples discussed in this review also illustrate how good design and thorough characterization are valuable in the development of delivery systems.

We have brought here an overview of the properties of polymeric nanoparticles that influence brain delivery, illustrated with examples that point out the importance of comprehensive particle characterization. Particularly in view of targeting ligands, much remains to be explored and understood concerning avidity, ligand density, and protein corona formation. Importantly, no property can singly outline the brain delivery capabilities of a drug delivery system, making every particle unique. Valuable trends can, however, be extrapolated, unraveling important design criteria for novel nanocarriers that can reliably gain access to the brain.

## Figures and Tables

**Figure 1 pharmaceutics-13-02045-f001:**
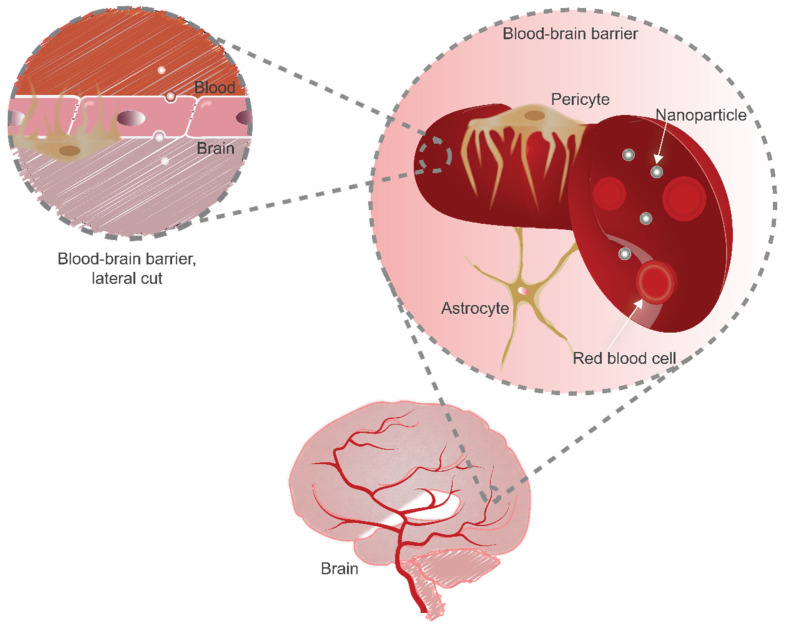
Blood–brain barrier representation. The brain capillaries irrigate the brain parenchyma, and their structure is composed of specialized endothelial cells as well as pericytes and astrocytes. The endothelial cells display apical–basal polarity, and tight junctions separate the polarized membranes. To reach the brain, NPs will need to interact with the apical membrane, be internalized by the endothelium, and undergo vesicular trafficking and exocytosis.

**Figure 2 pharmaceutics-13-02045-f002:**
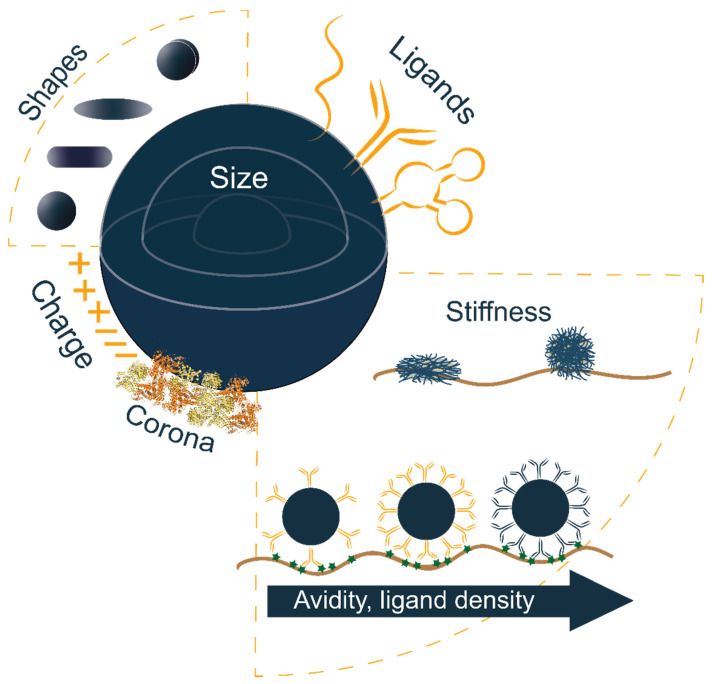
Schematic representation of a polymeric drug delivery system and its tunable properties. This summarizes the main properties of polymer-based nanocarriers for controlling the interaction with biological systems and improving delivery efficacy. Human serum albumin (HSA) protein structure in “Corona” was adapted from the Protein Data Bank (http://www.pdb.org, accessed on 19 October 2021) PDB ID 1AO6.

**Figure 3 pharmaceutics-13-02045-f003:**
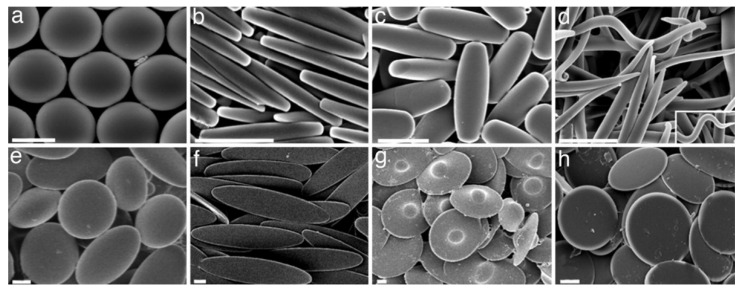
Representative micrographs of anisotropic and non-anisotropic polystyrene nanoparticles prepared by film stretching. Nanoparticles shape are (**a**) spherical, (**b**) rectangular discoidal, (**c**) rod-shaped, (**d**) worm-like, (**e**) oblate ellipsoidal, (**f**) elliptical discoidal, (**g**) UFO-like, and (**h**) circular discoidal. Scale bars are 2 µm. Adapted with permission from Reference [37], National Academy of Sciences, 2007.

**Figure 4 pharmaceutics-13-02045-f004:**
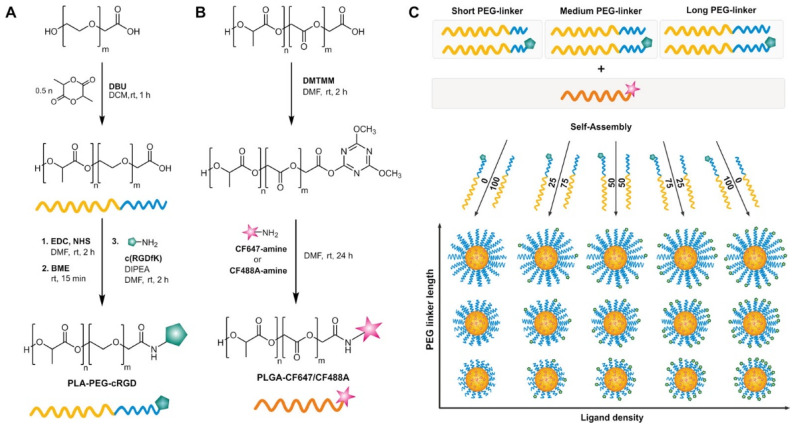
Strategy for the preparation of polymeric NPs with different linker length and ligand density. (**A**) Synthesis of poly(lactic acid)–poly(ethylene glycol) copolymer with carboxyl end (PLA-PEG-COOH) and different lengths of PEG (2, 3.5 and 5 kDa) and conjugation of the targeting moiety (Cyclo(RGDfK)) by EDC/NHS chemistry. (**B**) Fluorescent-labeling with CF™647/CF™488A amine dyes to carboxylic acid-terminated PLGA using 4-(4,6-dimethoxy-1,3,5-triazin-2-yl)-4-methylmorpholinium chloride (DMTMM). (**C**) Preparation of polymeric NPs with different PEG linker lengths and ligand densities. EDC: N-(3-dimethylaminopropyl)-N′-ethylcarbodiimide, NHS: N-hydroxysuccinimide, DBU: 1,8-Diazabicyclo[5.4.0]undec-7-ene, BME: beta-mercaptoethanol, DCM: dichloromethane, DMF: dimethylformamide, DIPEA: N,N-diisopropylethylamine. Adapted with permission from Reference [63], American Chemical Society, 2019.

**Figure 5 pharmaceutics-13-02045-f005:**
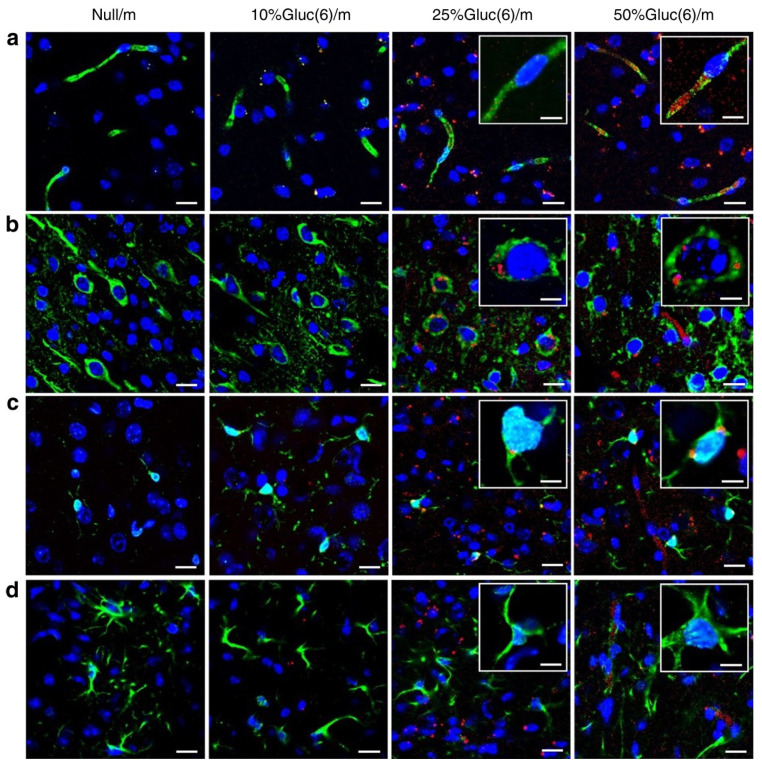
Interaction between C6-position glucose-functionalized polymeric micelles containing different glucose densities and the mouse brain. Sections were immunolabelled with (**a**) anti-PECAM1 antibody for brain capillary endothelial cells, (**b**) anti-Tuj1 antibody for neurons, (**c**) anti-Iba1 antibody for microglia, and (**d**) anti-GFAP antibody for astrocytes (green); DAPI (blue) for nuclei and micelles are labelled with fluorescent dye Cy5 (red). Scale bar: 20 μm (10 μm in insets). Adapted with permission from Reference [12], Springer Nature, 2017.

**Figure 6 pharmaceutics-13-02045-f006:**
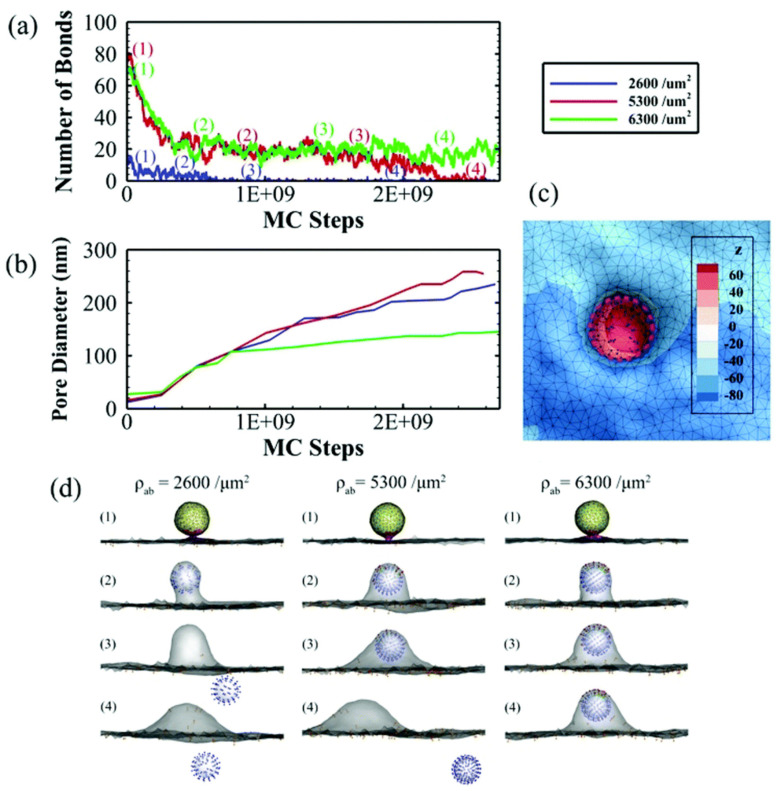
Simulations of the exocytosis of ligand-functionalized nanoparticles with three different densities of ligands (ρ_ab_ = 2600, 5300 and 6300 μm^−2^) on the receptor-mediated transcytosis across polarized cells. (**a**) Number of bonds formed between ligand and receptor. (**b**) Fusion pore formation with diameter progression. (**c**) Bottom view of the membrane and pore diameter. (**d**) Evolution of the vesicle interaction with membrane and fusion pore formation followed by detachment from the membrane for sufficiently low ligand densities. Nanoparticles are considered rigid spheres and ligands are considered pH independent. The last condition guarantees that the number of bonds and receptors that is at the end of endocytosis is the same as the number that is at the beginning of exocytosis. Adapted with permission from Reference [65], Royal Society of Chemistry, 2019.

**Table 1 pharmaceutics-13-02045-t001:** Summary of the effects of polymeric NPs properties in brain delivery and NPs properties advantages and limitations in application.

NP Property	Effect on Brain Delivery	Advantages	Limitations
Size	Smaller sizes tend to benefit transport across the BBB.	NP size can be controlled through different methods.	Preparation of smaller polymeric NPs is still challenging, and size population homogeneity is often depreciated.
	Brain accumulation is shown to be higher for smaller NPs.	Size control may improve brain accumulation.	Loading of molecules may be low for small sized NPs.
			Smaller NPs are more prone to clearance by the kidneys, while bigger NPs tend to be cleared by the spleen.
Shape	Specific interactions may be favored in ligand-functionalized NPs.	Specific NP shapes may increase cell adhesion, e.g., rods.	Synthesis methods are not yet straightforward or broadly applicable.
			May reduce internalization by cells because more energy is required for wrapping.
Stiffness	Stiffer NPs usually display increased uptake but are not necessarily more transcytosed.	Softer particles display reduced protein adsorption.	Uptake of softer NPs by cells is most likely reduced, which may limit treatment efficacy.
	NP brain accumulation is dependent on stiffness.	Variety of methodologies available to control stiffness.	The influence of other physicochemical properties might not allow setting an unequivocal threshold for an extensive range of particles.
	Effects are highly dependent on the range of stiffness (often Young’s module).	Softer particles may be used to evade the MPS and enhance brain accumulation.	Stiffness modulation may not be trivial to all systems and all stiffnesses ranges.
			NPs stiffness might not be homogeneous through the particle.
Charge	Negatively charged surface of endothelial cells favors interaction with positively charged particles.	Control over surface charge may be used to diminish accumulation in endothelial cells and improve transport across the BBB.	Higher uptake does not necessarily lead to higher transcytosis, as positively charged particles might remain trapped in the endothelial cells to a larger extent.
	Positively charged NPs show higher uptake but lower transport across cells barriers compared with negatively charged NPs.		Positively charged particles may induce toxicity, increase ROS, and affect BBB integrity.
Ligands	May increase targetability	Improves specificity.	Targeting ability may be limited by protein corona formation.
	May increase transcytosis but also depends on ligand density and affinity.	Versatility of conjugation techniques.	Unspecific-site functionalization strategies may reduce receptor-ligand interaction; additionally, populations heterogeneity is expected.
		Variety of ligands.	Specific targets must be previously identified.
		Use of multiple ligands for dual- or multi-targeting.	Size increases due to functionalization.
			Some ligands are costly and hard to produce and purify, e.g., antibodies.
Avidity	Transcytosis is boosted by tuning ligand density and avoiding enduring attachment of particles to the endothelial cells membrane.	Control of the endocytosis, sorting, and exocytosis in endothelial cells.	Controlled ligand density is not trivial.
	Avidity regulates the levels and location of NPs in the brain.	Improvement in the therapeutic index of drug delivery systems.	Engineering and production of ligands with different affinities is complex.
		Enhance uptake by target cell.	
Corona	Presence of specific proteins, e.g., Apo E, may enhance NPs transport across the BBB and accumulation in the brain.	May reduce particle toxicity.	Alters size, shape, and surface properties.
		May improve particle targetability.	May hamper targetability of ligand-functionalized NPs by masking the ligands.
			Affects predictability of NPs–biological environment interaction.

## Data Availability

Not applicable.

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
