# Peer review of "Polymeric Nanoparticles Properties and Brain Delivery"

_pharmaceutics, 2021, doi:10.3390/pharmaceutics13122045_

Round 1
Reviewer 1 Report
The manuscript describes the latest developments in the synthesis and use of polymeric nanoparticles for brain delivery therapeutic strategies. The manuscript was carefully prepared, logically organized and is easy to follow. The introduction is comprehensive and contains all the necessary information. The topic is interesting and falls in the readership of Pharmaceutics.
Some specific comments that aim to improve the manuscript are listed below:
Additional justification about the need for this review given other recent reviews in this field is recommended. Perhaps specific differences between this review and the previous ones would help.
I would suggest adding a comparison among the targeting ligands. The advantages and disadvantages of their use and the development of delivery systems would be of interest. Hence, conclusions could be focused more on the transfer of these technologies in delivery devices.
What are the important trends and research gaps that the papers mentioned are endeavouring to address? Significant results in the field should be highlighted and discussed accordingly. This sort of commentary is not needed for every instance where the authors have summarized key references but should be done more than what has already been done so that the manuscript is more a review than a summary list of recent papers.
Author Response
We thank the reviewers for their rapid and careful evaluation of the manuscript and we have revised the manuscript accordingly. We hope that with the current corrections we have incorporated, the manuscript is now acceptable for publication.
Reviewers’ responses are pasted in black. Authors’ responses are in blue. Changes in the manuscript have also been marked in blue.
Reviewer 1
The manuscript describes the latest developments in the synthesis and use of polymeric nanoparticles for brain delivery therapeutic strategies. The manuscript was carefully prepared, logically organized and is easy to follow. The introduction is comprehensive and contains all the necessary information. The topic is interesting and falls in the readership of Pharmaceutics.
Some specific comments that aim to improve the manuscript are listed below:
1. Additional justification about the need for this review given other recent reviews in this field is recommended. Perhaps specific differences between this review and the previous ones would help.
Response:
We thank the reviewer for pointing this out, and we added a more detailed description to the abstract section which identifies the outstanding topics in the review as compared to previous literature. We highlight the discussion on stiffness and ligands and ligands-functionalization properties and how it has an effect on avidity and ultimately particles interaction with the biological environments. The following text has been included:
“Here, we discuss how brain delivery by NPs can benefit from careful design of the NP proper-ties. Properties such as size, charge, shape and ligand-functionalization are commonly addressed in the literature, however, properties like ligand density, linker length, avidity, protein corona and stiffness are insufficiently discussed. This is unfortunate, since they present great value against multiple barriers encountered by the NPs before reaching the brain, particularly the BBB. We further highlight important examples utilizing targeting ligands and how functionalization parameters, e.g., ligand density and ligand properties, can affect the success of the nano-based delivery system.”
2. I would suggest adding a comparison among the targeting ligands. The advantages and disadvantages of their use and the development of delivery systems would be of interest. Hence, conclusions could be focused more on the transfer of these technologies in delivery devices.
Response:
We have prepared a table summarizing the physicochemical properties of NPs and their influences on brain delivery (Table 1, end of Section 5, page 17), including the influence of targeting ligands from a general perspective, as the effects of targeting ligands is difficult to predict.
3. What are the important trends and research gaps that the papers mentioned are endeavouring to address? Significant results in the field should be highlighted and discussed accordingly. This sort of commentary is not needed for every instance where the authors have summarized key references but should be done more than what has already been done so that the manuscript is more a review than a summary list of recent papers.
Response:
We have deepened the discussion at several key points regarding the relation of NP properties on brain delivery to highlight the significance of these contributions. The textual changes are shown below.
(3. Nanoparticle shape, page 5)
NP anisotropy influences particle adhesion and avidity. Studies indicate that ligand functionalized NPs that induce specific interactions, may profit from anisotropy and enhanced targeting may ensue [15,31]. Still, the impact of different shapes of polymeric NPs in brain delivery requires more investigation, which is currently limited by the lack of well-established and broadly applicable methodologies to prepare such NPs.
(4. Nanoparticle stiffness, page 6)
Employing a filter-free BBB model and p(NIPMAM) nanogels with different crosslinking densities, Zuhorn and coworkers showed that lower stiffness nanogels (<200 kPa) promote transcytosis even though endocytosis is not promoted [44].
(4. Nanoparticle stiffness, page 6)
The influence of stiffness on transcytosis and brain accumulation adds a parameter to the design of NPs that may be evaluated for polymeric NPs. Comparison should always observe the stiffness range, often described by the particles’ Young’s moduli, and not simply as softer or stiffer.
(5.3.1. Nanoparticle surface characteristics, Targeting ligands, Glucose and glucose derivatives – Glucose transporters, page 9)
Kataoka and coworkers described above addresses in a comprehensive manner how relevant control over ligand density is for NP-mediated brain delivery, distinguishing association levels of micelles to different components of the NVU.
(5.2. Nanoparticle surface characteristics, Ligand density and linker length, page 7)
Ligand density has a substantial impact on the avidity of the NPs, which ultimately affects their uptake, intracellular trafficking and transcytosis, which is discussed to greater extent in Section 5.4 Avidity.
(5.4. Nanoparticle surface characteristics, Avidity, page 14)
The effect of particle avidity on associating with the sorting tubules during intracellular trafficking sheds light on a remarkable sorting mechanism and factors that may control it. It not only reinforces the importance of NP design to improve brain delivery, but also points out that controlled ligand functionalization is beneficial for targeted drug delivery.
(6. Concluding remarks, page 19)
The examples discussed in this review also illustrate how good design and through characterization are valuable on the development of delivery systems.
Reviewer 2 Report
The manuscript was well written and only several minor issues are required to be addressed before becoming acceptable.
1, The main contents of the manuscript are focused on the brain delivery-related properties, but several subsections contain contents that have nothing to do with the topic. Check Line 462, line 495, etc. If the gene delivery and tumors are brain delivery relevant, please provide more details.
2, Section 5.2, the summary on the link length was missed.
3, If possible, please summarize effects of all the parameters on the brain delivery in one table or fig.
4, Line 224, “HUVEC cells”. Please give the full name of HUVEC, and it includes the word cell already.
5, Line 83, “the majority of all nps”. Delete the word all.
6, Abbreviation use issue. There is no abbreviation of NPs in the text. Check other key nouns. Also, many chemical molecule names were used only once, and the abbreviations are not required.
7, It would be better if the side effects of each parameter on the brain delivery can be discussed.
8, Fig2, Use plural form of shape vs ligands.
9, Line 445, RGD is for Arg-Gly-Asp.
Author Response
We thank the reviewers for their rapid and careful evaluation of the manuscript and we have revised the manuscript accordingly. We hope that with the current corrections we have incorporated, the manuscript is now acceptable for publication.
Changes in the manuscript have been marked in blue.
Reviewer 2
The manuscript was well written and only several minor issues are required to be addressed before becoming acceptable.
1. The main contents of the manuscript are focused on the brain delivery-related properties, but several subsections contain contents that have nothing to do with the topic. Check Line 462, line 495, etc. If the gene delivery and tumors are brain delivery relevant, please provide more details.
Response:
We thank the reviewer for bringing up this pertinent point. The review was carefully tailored to address relevant topics in brain delivery which comprises both direct studies on particles properties and delivery systems applications, e.g., treatment of glioblastoma, Alzheimer’s and Parkinson’s disease. Those systems carry not only drug molecules, but also other types of active pharmaceutical ingredients such as DNA and siRNA. Also, dual targeting systems are described to illustrate benefits of multivalent targeting. To assure clarity to a broader audience we have implemented several additions to the text.
Changes to the text:
(2. Nanoparticle size, page 4)
The highest cellular internalization in Caco-2, human colorectal adenocarcinoma cells, and Madin-Darby Canine Kidney (MDCK) cells were observed for 100 nm NPs, however, this was only significant for TPGS coated NPs in the MDCK cells. Even though Caco-2 and MDCK cells are not brain endothelial cells, they are epithelial cells that also form cell barriers and a parallel for the particles behavior in polarized brain endothelial cell barriers can be done to an extent.
(5.3.3. Nanoparticle surface characteristics, Peptides, RGD peptide, page 12)
In a dual targeting approach aimed at superior targeting ability and delivery to tumor ne-ovasculature and tumor cells, PEG-PCL nanoparticles were functionalized with RGD peptide and interleukin-13 peptide [123]. Ex vivo imaging of treated mice indicated great accumulation of the dual functionalized NPs in the tumor site in the brain, also deeper penetration in tumor spheroids was observed for the dual functionalized particles.
(5.3.3. Nanoparticle surface characteristics, Peptides, RVG peptide, page 13)
RVG peptides have also been reported as potential carriers for gene therapies to the brain [135, 136].
(5.4. Nanoparticle surface characteristics, Avidity, page 14)
The simulation aims to predict the behavior of NPs with different avidities through receptor-mediated transcytosis, an essential process on the transport of NPs across the BBB to reach the brain.
2. Section 5.2, the summary on the link length was missed.
Response:
We thank the reviewer for noticing this and have added a short summary at the end.
“Ligand density has a substantial impact on the avidity of the NPs, which ultimately affects their uptake, intracellular trafficking and transcytosis, which is discussed to greater extent in Section 5.4 Avidity.”
3. If possible, please summarize effects of all the parameters on the brain delivery in one table or fig.
Response:
A table (Table 1) summarizing the effects of polymeric NPs properties in brain delivery and NPs properties advantages and limitations in application was added to the manuscript. The table can be found at the end of Section 5.
4. Line 224, “HUVEC cells”. Please give the full name of HUVEC, and it includes the word cell already.
Response:
Changes to the text (Nanoparticle surface characteristics, Surface charge):
The endothelial cells of the BBB have a higher density of negative charges due to the presence of proteoglycans, as compared to blood components and human umbilical vein endothelial cells (HUVEC) [59].
5. Line 83, “the majority of all nps”. Delete the word all.
Response:
We have made the according change.
6. Abbreviation use issue. There is no abbreviation of NPs in the text. Check other key nouns. Also, many chemical molecule names were used only once, and the abbreviations are not required.
Response:
The definition of the abbreviation NPs is mentioned in the abstract section, and to improve readability it has also been included in the introduction section. We have revised the other abbreviations according to the reviewer’s suggestion.
7. It would be better if the side effects of each parameter on the brain delivery can be discussed.
Response:
A table (Table 1) summarizing the effects of polymeric NP properties in brain delivery and NP properties advantages and limitations in application was added to the manuscript. The table can be found at the end of section 5.
8. Fig2, Use plural form of shape vs ligands.
Response:
We appreciate the suggestion and have made the change.
9. Line 445, RGD is for Arg-Gly-Asp.
Response:
We thank the reviewer for noticing this mistake, we corrected the 3-letters code amino acid abbreviation.
Reviewer 3 Report
This manuscript reviews various articles on the use of polymeric nanoparticles as a means of brain delivery and the effects of each physical property for brain transport, such as the size, shape, stiffness, and surface properties of nanoparticles. The manuscript is well written and structured, and each category can be a good reference for other researchers. The big question, however, is that there are several similar review articles. Therefore, it is necessary to insert more articles about the latest technology to brain delivery using polymer nanoparticles. Also, a few figures are included in this review, so more figures are needed to help the reader understand.
Author Response
We thank the reviewers for their rapid and careful evaluation of the manuscript and we have revised the manuscript accordingly. We hope that with the current corrections we have incorporated, the manuscript is now acceptable for publication.
Changes in the manuscript have been marked in blue.
Reviewer 3
This manuscript reviews various articles on the use of polymeric nanoparticles as a means of brain delivery and the effects of each physical property for brain transport, such as the size, shape, stiffness, and surface properties of nanoparticles. The manuscript is well written and structured, and each category can be a good reference for other researchers. The big question, however, is that there are several similar review articles. Therefore, it is necessary to insert more articles about the latest technology to brain delivery using polymer nanoparticles. Also, a few figures are included in this review, so more figures are needed to help the reader understand.
Response:
We have added to the abstract a clearer description of what the review brings that makes it stand out from the other previous reviews.
The review covers the recent literature on polymeric nanoparticles focusing in brain delivery, in particularly, studies that allow to extrapolate on properties effect on the interaction between nanoparticles and biological environments. The majority of the cited literature is from the years 2018-2021 with about half from 2020-2021. The review addresses the latest technologies in terms of NP’s properties modulation benefits to particle design for brain delivery and they are the effect of particles avidity, especially ligand density and ligand affinity addressing the association with sorting tubules regulation on transcytosis besides effect of stiffness and ligand-decorated nanoparticles. We have also extended the discussion at key points on the relation of NP properties and brain delivery to highlight the significance of the contributions, as also indicated by Reviewer 1. A table summarizing NP properties effects was also added and will contribute more to the reader’s understanding (see Table 1 in the manuscript, end of Section 5).
Changes to the text have been indicated in the response to Reviewer 1 (comments 1 & 3)
(Abstract)
“Here, we discuss how brain delivery by NPs can benefit from careful design of the NP proper-ties. Properties such as size, charge, shape and ligand-functionalization are commonly addressed in the literature, however, properties like ligand density, linker length, avidity, protein corona and stiffness are insufficiently discussed. This is unfortunate, since they present great value against multiple barriers encountered by the NPs before reaching the brain, particularly the BBB. We further highlight important examples utilizing targeting ligands and how functionalization parameters, e.g., ligand density and ligand properties, can affect the success of the nano-based delivery system.”
(3. Nanoparticle shape, page 5)
NP anisotropy influences particle adhesion and avidity. Studies indicate that ligand functionalized NPs that induce specific interactions, may profit from anisotropy and enhanced targeting may ensue [15,31]. Still, the impact of different shapes of polymeric NPs in brain delivery requires more investigation, which is currently limited by the lack of well-established and broadly applicable methodologies to prepare such NPs.
(4. Nanoparticle stiffness, page 6)
Employing a filter-free BBB model and p(NIPMAM) nanogels with different crosslinking densities, Zuhorn and coworkers showed that lower stiffness nanogels (<200 kPa) promote transcytosis even though endocytosis is not promoted [44].
(4. Nanoparticle stiffness, page 6)
The influence of stiffness on transcytosis and brain accumulation adds a parameter to the design of NPs that may be evaluated for polymeric NPs. Comparison should always observe the stiffness range, often described by the particles’ Young’s moduli, and not simply as softer or stiffer.
(5.3.1. Nanoparticle surface characteristics, Targeting ligands, Glucose and glucose derivatives – Glucose transporters, page 9)
Kataoka and coworkers described above addresses in a comprehensive manner how relevant control over ligand density is for NP-mediated brain delivery, distinguishing association levels of micelles to different components of the NVU.
(5.2. Nanoparticle surface characteristics, Ligand density and linker length, page 7)
Ligand density has a substantial impact on the avidity of the NPs, which ultimately affects their uptake, intracellular trafficking and transcytosis, which is discussed to greater extent in Section 5.4 Avidity.
(5.4. Nanoparticle surface characteristics, Avidity, page 14)
The effect of particle avidity on associating with the sorting tubules during intracellular trafficking sheds light on a remarkable sorting mechanism and factors that may control it. It not only reinforces the importance of NP design to improve brain delivery, but also points out that controlled ligand functionalization is beneficial for targeted drug delivery.
(6. Concluding remarks, page 19)
The examples discussed in this review also illustrate how good design and through characterization are valuable on the development of delivery systems.
Round 2
Reviewer 1 Report
The manuscript is ready for publication.